# FSTL1 promotes dendritic cell pyroptosis and immunosuppression in sepsis by inhibiting STING autophagy

Qiong Li[1◉], Hua Ling[2], Jingyi Li[3], Jinru Li[2], Wentao Duan[2], Lijun Guo[4]*, Xingui Dai[1,2]*

1 Department of Critical Care Medicine, First Clinical Medical College of Jinan University, The First People's Hospital of Chenzhou, Chenzhou, China, 2 Department of Critical Care Medicine, Affiliated Chenzhou Hospital (the First People's Hospital of Chenzhou), Southern Medical University, Chenzhou, China, 3 Xu Zhou Medical University College of Pharmacy, Xuzhou, China, 4 Department of Neurology, Affiliated Hospital of Xiangnan University, Chenzhou, China

◉ This author contributed equally to this work.
* dyce@2008.sina.com (XD); guolijun54@126.com (LG)

## Abstract

### Background

Sepsis-induced immunosuppression, characterized by dendritic cell (DC) depletion, correlates with poor outcomes. The glycoprotein FSTL1 is elevated in sepsis, but its contribution to DC pyroptosis and subsequent immune dysfunction remains unknown.

### Methods

We utilized *in vitro* models with splenic DCs and mouse dendritic cell line DC2.4, alongside a murine cecal ligation and puncture (cecal ligation and puncture) sepsis model. The roles of STING and autophagy were probed using specific modulators (STING agonist DMXAA; STING specific inhibitor C-176; autophagy inhibitor 3-methyladenine). Pyroptosis was assessed by quantifying cleaved CASP1 and GSDMD-N via Western blotting and flow cytometry. STING pathway activation and autophagic flux were evaluated by detecting protein phosphorylation (p-STING, p-TBK1) and key markers (LC3B, P62) through Western blotting, immunofluorescence, and co-immunoprecipitation. DC-mediated T-cell responses were determined by proliferation assays and cytokine secretion analysis.

### Results

FSTL1 was found elevated and correlated with DC pyroptosis *in vitro* and in septic mice. Mechanistically, FSTL1 inhibited the autophagic degradation of STING, leading to its accumulation and subsequent activation. Consequently, this impaired T-cell priming capacity and resulted in immunosuppression *in vivo*. Inhibition of STING attenuated FSTL1-induced pyroptosis, restored DC-mediated T-cell activation, and

**Data availability statement:** All relevant data are within the manuscript and its Supporting information files.

**Funding:** This study was financially supported by the Health Commission of Hunan Province in the form of grants received by LG (C2019136) and QL (D202317018022). This study was also financially supported by the Education Department of Hunan Province in the form of a grant (17C1477) received by LG. The funders had no role in study design, data collection and analysis, decision to publish, or preparation of the manuscript.

**Competing interests:** The authors have declared that no competing interests exist.

ameliorated immunosuppression. In murine septic models, FSTL1 exacerbated multiple organ injury and increased mortality, effects that were reversed by STING inhibition.

## Conclusion

Our findings demonstrate that FSTL1 correlates with impaired STING autophagic degradation and DC pyroptosis, suggesting a potential pathway contributing to septic immune dysfunction.

## Introduction

According to the latest international definition, sepsis is characterized by a dysregulated host response to infection that leads to life-threatening organ dysfunction. It remains a serious global health issue, imposing a substantial medical, social, and economic burden [1]. Immune dysregulation is central to the pathogenesis of sepsis [2–4]. In particular, DC depletion has been closely associated with impaired immune function, disease severity, poor prognosis, and high mortality [5]. However, the molecular mechanisms driving DC death in sepsis are not fully understood.

Pyroptosis is a highly inflammatory form of programmed cell death, primarily mediated by gasdermin D (GSDMD). Unlike apoptosis, pyroptosis is characterized by the massive release of inflammatory mediators such as IL-1β and IL-18, which critically amplify immune responses [6]. This process is initiated through caspase-1-dependent canonical or caspase-4/5/11-dependent non-canonical pathways, leading to cleavage of GSDMD, pore formation in the plasma membrane, cell rupture, and proinflammatory cytokine release [7,8]. In the context of sepsis, pyroptosis serves a protective role by eliminating intracellular pathogens and restricting bacterial replication, partly through gasdermin-induced pores in bacterial membranes. However, when excessively activated, it propagates uncontrolled systemic inflammation, widespread immunocyte death, multi-organ damage, and secondary immunosuppression, contributing significantly to poor clinical outcomes [9]. Furthermore, pyroptosis has been implicated in sepsis-related organ injuries such as disseminated intravascular coagulation, acute kidney injury, acute lung injury, and cardiac dysfunction [10]. These findings highlight the pivotal yet complex role of pyroptosis in sepsis by not only mediating innate immune defense but also driving pathological inflammation and organ failure, underscoring its potential as a therapeutic target. The pyroptotic death of DCs, the most potent antigen-presenting cells, would critically impair the initiation of adaptive immune responses, thus directly contributing to the state of immunosuppression observed in sepsis. Despite this, whether and how pyroptosis contributes to the depletion of dendritic cells, which are key orchestrators of adaptive immunity, during sepsis remains an area of active investigation.

Follistatin-like protein 1 (FSTL1) is a secreted glycoprotein implicated in inflammatory processes. It enhances the production of proinflammatory cytokines and chemokines, and promotes inflammatory responses in various disease models [11]. FSTL1

expression is elevated under inflammatory conditions and decreases upon treatment, suggesting its potential as a bio-marker and therapeutic target in inflammation-driven diseases [12]. Recent studies have indicated that FSTL1 also plays a critical role in the immunosuppressive microenvironment. Glioma stem cells (GSCs) secreted FSTL1 was significantly upregulated in tumors and enhanced immunosuppressive function by promoting M2 polarization of tumor-associated macrophages (TAMs) [13]. Other studies have reported elevated FSTL1 levels in patients with bacterial sepsis, where it promotes NLRP3 inflammasome-dependent IL-1β secretion in monocytes and macrophages [14]. While FSTL1 is known to promote pro-inflammatory responses in certain cells like macrophages, its specific role in antigen-presenting cells such as DCs, particularly in the context of sepsis-induced immunosuppression, remains a critical unanswered question. However, its role in DC pyroptosis and sepsis-induced immunosuppression remains unexplored.

Based on these observations, we hypothesized that FSTL1 triggers pyroptosis in DCs and contributes to immune dysfunction in sepsis. In this study, we combined *in vitro* and *in vivo* approaches to investigate whether FSTL1 promotes DC pyroptosis via the STING pathway and whether its inhibition can restore immune function. Our findings reveal a novel mechanism through which FSTL1 impairs autophagic degradation of STING, thereby amplifying pyroptotic cell death and immunosuppression. This study not only deepens our understanding of septic immunopathology but also suggests FSTL1 as a potential biomarker and therapeutic target for sepsis.

## Results

### Elevated serum FSTL1 correlates with an immunosuppressive phenotype in septic mice

To investigate the role of FSTL1 in sepsis-induced immunosuppression, we evaluated serum levels of FSTL1, transforming growth factor-beta (TGF-β), IL-2, IL-4, and IL-10 in a murine sepsis model at different time points (0, 1, 2, and 3 days post-induction). We observed that FSTL1 levels increased progressively with sepsis progression. This elevation positively correlated with levels of the anti-inflammatory cytokines TGF-β, IL-4, and IL-10, and negatively correlated with the pro-inflammatory cytokine IL-2 (Fig 1A–1E). Furthermore, immunophenotypic analysis of peripheral blood mononuclear cells (PBMCs) revealed a gradual decline in the percentages of CD3$^+$ and CD4$^+$T cells, accompanied by a rising proportion of CD4$^+$Foxp3$^+$regulatory T cells (Tregs) over the course of sepsis (Fig 1F–1H).

Collectively, these data show that rising FSTL1 levels parallel the development of T-cell exhaustion and an immunosuppressive cytokine milieu in sepsis, suggesting a potential link between FSTL1 and septic immune dysfunction.

### FSTL1 directly promotes DC pyroptosis and impairs T-cell priming capacity

To determine whether FSTL1 contributes to DC pyroptosis during sepsis, we first examined DC death in the spleen at different time points (0, 1, 2, and 3 days) after CLP. Prior to assessing pyroptosis, we confirmed that the purity of isolated splenic CD11c$^+$DCs exceeded 90% (S1A Fig). We observed a time-dependent increase in Annexin V$^+$ and 7-AAD$^+$DC populations (S1B Fig), indicating elevated DC death concomitant with sepsis progression, and suggesting a potential positive correlation between FSTL1 and DC pyroptosis. To determine whether FSTL1 directly triggers this effect, we next examined its impact on splenic DCs *in vitro*. Splenic DCs were isolated and treated with LPS alone or in combination with FSTL1. Western blot analysis revealed that protein levels of cleaved CASP-1, GSDMD-N, and IL-1β were significantly higher in the FSTL1+LPS group compared to the LPS-only group (Fig 2A). Consistently, the rate of DC pyroptosis was also elevated upon FSTL1 co-treatment (Fig 2B). Moreover, the FSTL1+LPS group showed increased secretion of IL-1β, IL-18, HMGB1, and TNF-α in the cell culture supernatant relative to the LPS group (Fig 2C), further supporting a pro-pyroptotic role of FSTL1. To assess the functional impact of FSTL1-enhanced DC pyroptosis on T-cell immunity, we co-cultured differentially stimulated DCs with normal, untreated splenic CD4$^+$T cells. The FSTL1+LPS group resulted in significantly reduced proliferative activity of CD4$^+$T cells compared to the LPS group (Fig 2D). Additionally, co-culture supernatants from the FSTL1+LPS group contained lower levels of IL-2 and IFN-γ, but higher levels of IL-4 and

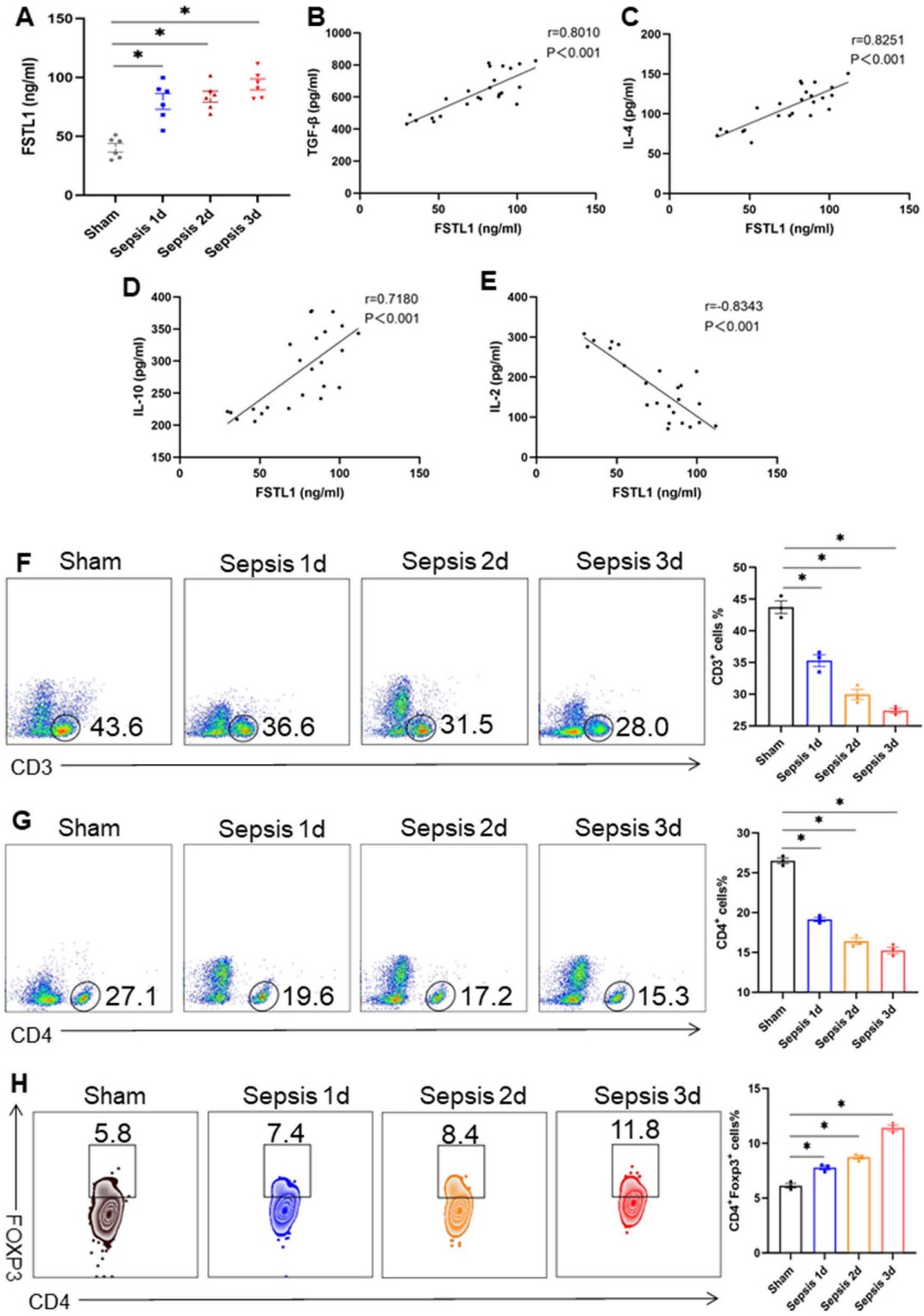

**Fig 1. Serum FSTL1 levels gradually increased during sepsis progression. (a-e)** C57BL/6 mice were subject to sham or CLP operations. Serum from septic mice was collected at different time points (0, 1, 2 and 3 **d)**, and levels of FSTL1, IL-2, IL-4, IL-10, and TGF-β were measured by ELISA (n = 6). **(f-h)** PBMCs were collected from sham or CLP-operated mice at different time points (0, 1, 2 and 3 **d)**, and the proportions of CD3+ T cells, CD4+T cells, and CD4+Foxp3+ T cells were analyzed by flow cytometry (n = 3). Data were represented as Mean ± SD, *$P < 0.05$.

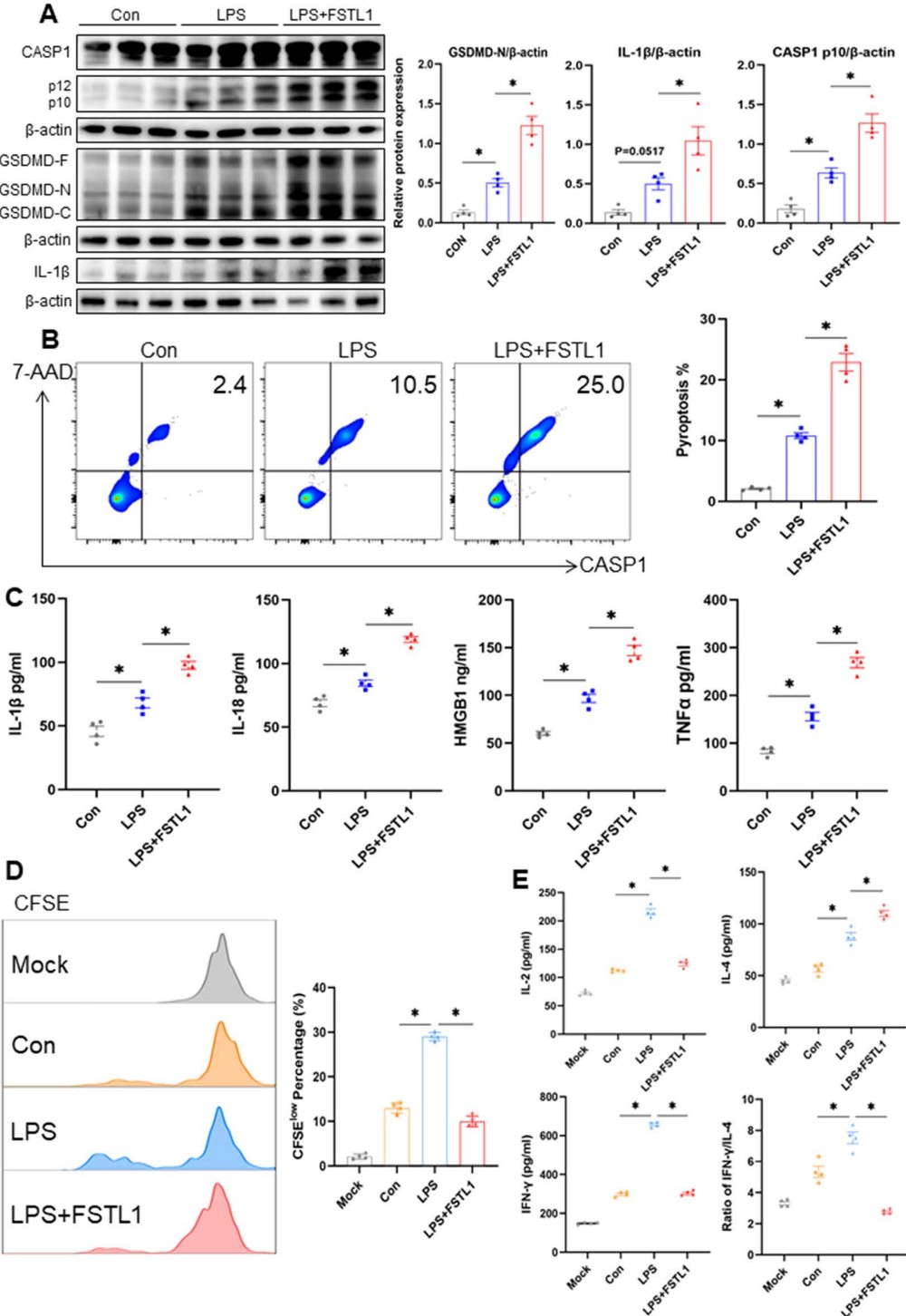

**Fig 2. FSTL1 promoted the pyroptosis of DCs. a-c** Splenic CD11+ DCs were isolated from C57BL/6 mice and exposed to the corresponding solvent (control), LPS (1ug/ml), or LPS+FSTL1 (10ng/ml) for 24h. The protein expression of cleaved caspase-1, GSDMD, IL-1β, and β-actin was detected by Western blot **(a)** (n = 3). The rate of DC pyroptosis was evaluated using flow cytometry **(b)** (n = 4). Level of IL-1β, IL-18, HMGB1 and TNF-α in the culture supernatants were evaluated by ELISA **(c)** (n = 4). **d-e** After stimulation, the DCs were co-cultured with T cells. The proliferation of CD4+ T cells co-cultured with DC in each group was measured based on CFSE assay **(d)** (n = 4). Cytokine levels (IL-2, IL-4, IFN-γ) and the IFN-γ/IL-4 ratio in the co-cultured supernatants were detected by ELISA **(e)** (n = 4). Data were represented as Mean ± SD, *P < 0.05.

a decreased IFN-γ/IL-4 ratio, indicating a shift toward Th2 polarization and impaired DC-mediated activation of adaptive immunity (Fig 2E).

Together, these results demonstrate that FSTL1 exacerbates DC pyroptosis during sepsis and attenuates their capacity to stimulate effective T-cell responses.

## FSTL1-induced DC pyroptosis is mediated by the STING signaling pathway

To investigate the role of STING signaling in pyroptosis, we employed DMXAA, a specific STING agonist [15], to stimulate the pathway *in vitro*. As a result, DMXAA treatment significantly increased the expression of GSDMD-N, cleaved CASP-1, and IL-1β, and elevated the rate of DC pyroptosis (Fig 3A, 3B), confirming that STING activation promotes pyroptotic cell death. We next investigated whether FSTL1 influences the STING pathway. Splenic DCs and DC2.4 cells were treated with either LPS alone or LPS combined with FSTL1. Protein analysis revealed that the ratios of pSTING/STING and pTBK1/TBK1 were markedly higher in the FSTL1＋LPS group compared to the LPS-only group (Fig 3C). Consistent with these findings, immunofluorescence staining showed enhanced pSTING intensity in FSTL1＋LPS-treated DC2.4 cells (Fig 3D). Furthermore, pronounced nuclear translocation of IRF3 was observed in the FSTL1＋LPS group (Fig 3E), indicating enhanced STING pathway activation. These results demonstrate that FSTL1 potentiates the activation of the STING signaling pathway, thereby aggravating DC pyroptosis during sepsis.

To further confirm the involvement of STING signaling in FSTL1-mediated pyroptosis, we inhibited STING activation using C-176 prior to FSTL1 and LPS stimulation [16]. Compared with the FSTL1＋LPS group, pretreatment with C-176 markedly reduced the ratio of pSTING/STING and pTBK1/TBK1, as well as the expression of GSDMD-N, cleaved CASP1, and IL-1β (Fig 4A). Consistent with these findings, C-176 treatment significantly suppressed the rate of DC pyroptosis induced by FSTL1 (Fig 4B). Furthermore, C-176 inhibited the release of IL-1β, IL-18, HMGB1, and TNF-α in the culture supernatant (Fig 4C). We next assessed whether STING inhibition could restore DC-mediated T-cell activation. The impaired CD4＋T-cell proliferative capacity observed in the FSTL1＋LPS group was partially rescued by C-176 treatment (Fig 4D). Accordingly, co-culture with DCs pretreated with FSTL1＋LPS＋C-176 enhanced the production of IL-2 and IFN-γ by CD4＋T cells, while reducing the secretion of IL-4 (Fig 4E). The increased IFN-γ/IL-4 ratio indicated a shift toward Th1-type immunity following STING inhibition (Fig 4E).

Collectively, these results demonstrate that blockade of STING signaling alleviates FSTL1-induced DC pyroptosis and restores adaptive immune function, underscoring the critical role of the STING pathway in FSTL1-mediated immunosuppression during sepsis.

## FSTL1 potentiates STING signaling by impairing its autophagic degradation

To elucidate the mechanism by which FSTL1 enhances STING signaling, we investigated whether FSTL1 modulates autophagic degradation of STING. As shown in Fig 5A, treatment with FSTL1＋LPS significantly increased the expression of P62 and decreased the LC3B-II/LC3B-I ratio in both splenic DCs and DC2.4 cells, accompanied by elevated STING protein levels. Concurrently, fluorescence imaging revealed enhanced FSTL1 intensity but reduced LC3B signal in DC2.4 cells treated with FSTL1＋LPS (Fig 5B), indicating impaired autophagic flux. To assess whether FSTL1 affects the autophagic degradation of STING, we examined the colocalization of STING with LC3B. Confocal microscopy showed significantly reduced STING-LC3B colocalization in the FSTL1＋LPS group compared to LPS alone (Fig 5C). Furthermore, co-IP experiments demonstrated that under LPS stimulation, LC3B interacted with both P62 and STING, while FSTL1 co-treatment significantly weakened the interaction between LC3B and STING (Fig 5D). To corroborate these findings, we used the autophagy inhibitor 3-MA to mimic FSTL1's effects. Consistent with the results above, 3-MA treatment suppressed STING degradation via autophagy, enhanced STING pathway activity, and exacerbated pyroptosis, as confirmed through Western blotting, immunofluorescence, and co-IP assays (Fig 5E-5F and S2 Fig).

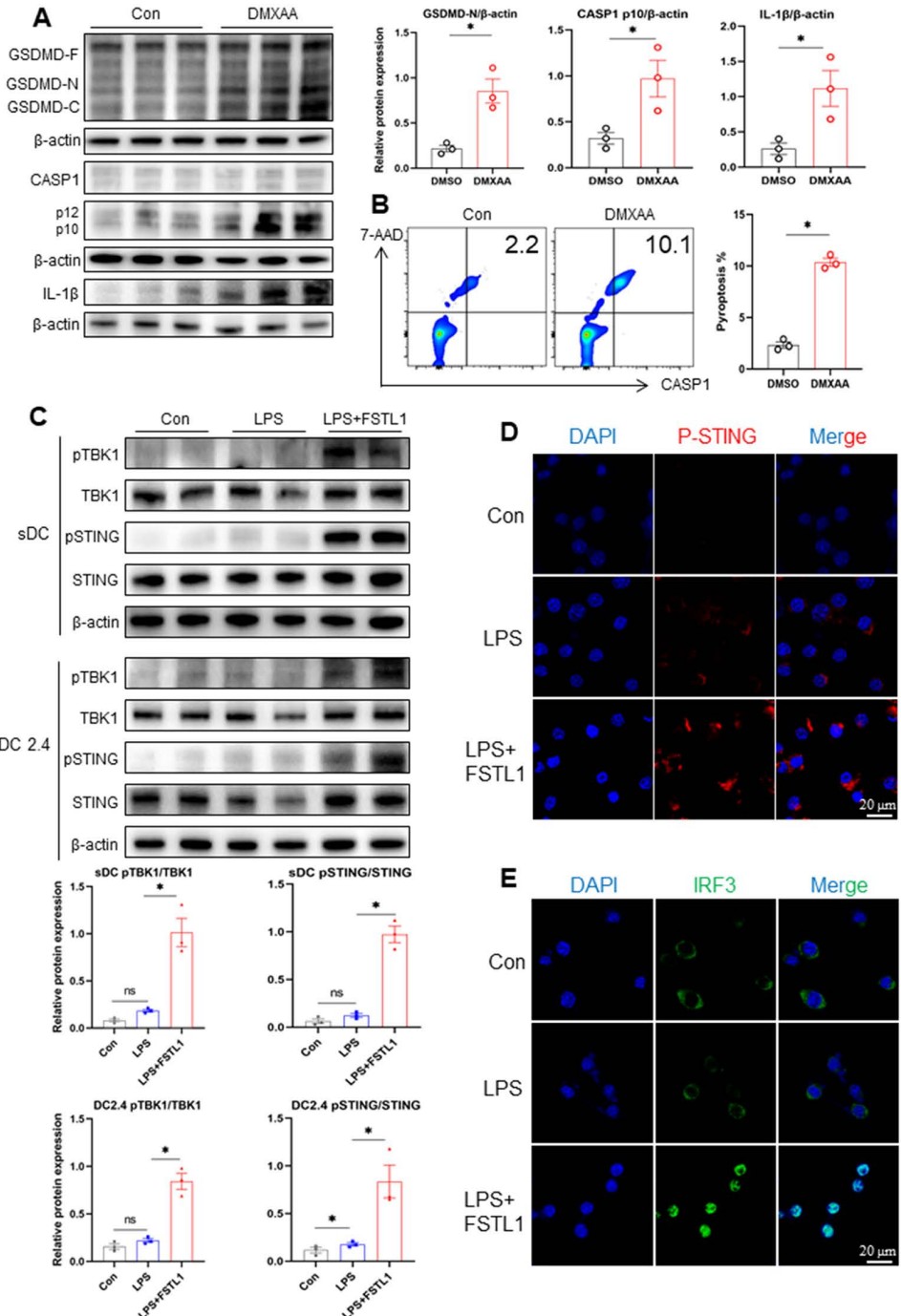

**Fig 3. FSTL1 promoted the STING-mediated pyroptosis activation. a-b** Splenic CD11+ DCs were isolated from C57BL/6 mice treated with DMSO (control) or DMXAA (1 μg/ml). The protein levels of cleaved caspase1, GSDMD, IL-1β, and β-actin detected by Western blotting **(a)** (n = 3). The pyroptosis rate of DCs was detected using flow cytometry **(b)** (n = 3). **c-e** Splenic CD11+ DC isolated from C57BL/6 mice and the DC2.4 cells cultured *in vitro* were exposed to PBS (control), LPS (1 μg/ml), and LPS+FSTL1 (10 ng/ml) for 24h. The protein levels of p-STING, STING, p-TBK1, TBK1, and β-actin in DCs and DC2.4 were detected by Western blotting **(c)** (n = 2). Representative confocal immunofluorescence images showing p-STING expression in DC2.4 cells under different treatments **(d)**. Representative confocal immunofluorescence images showing IRF3 expression in DC2.4 cells **(e)**. Data were represented as Mean±SD, *P<0.05.

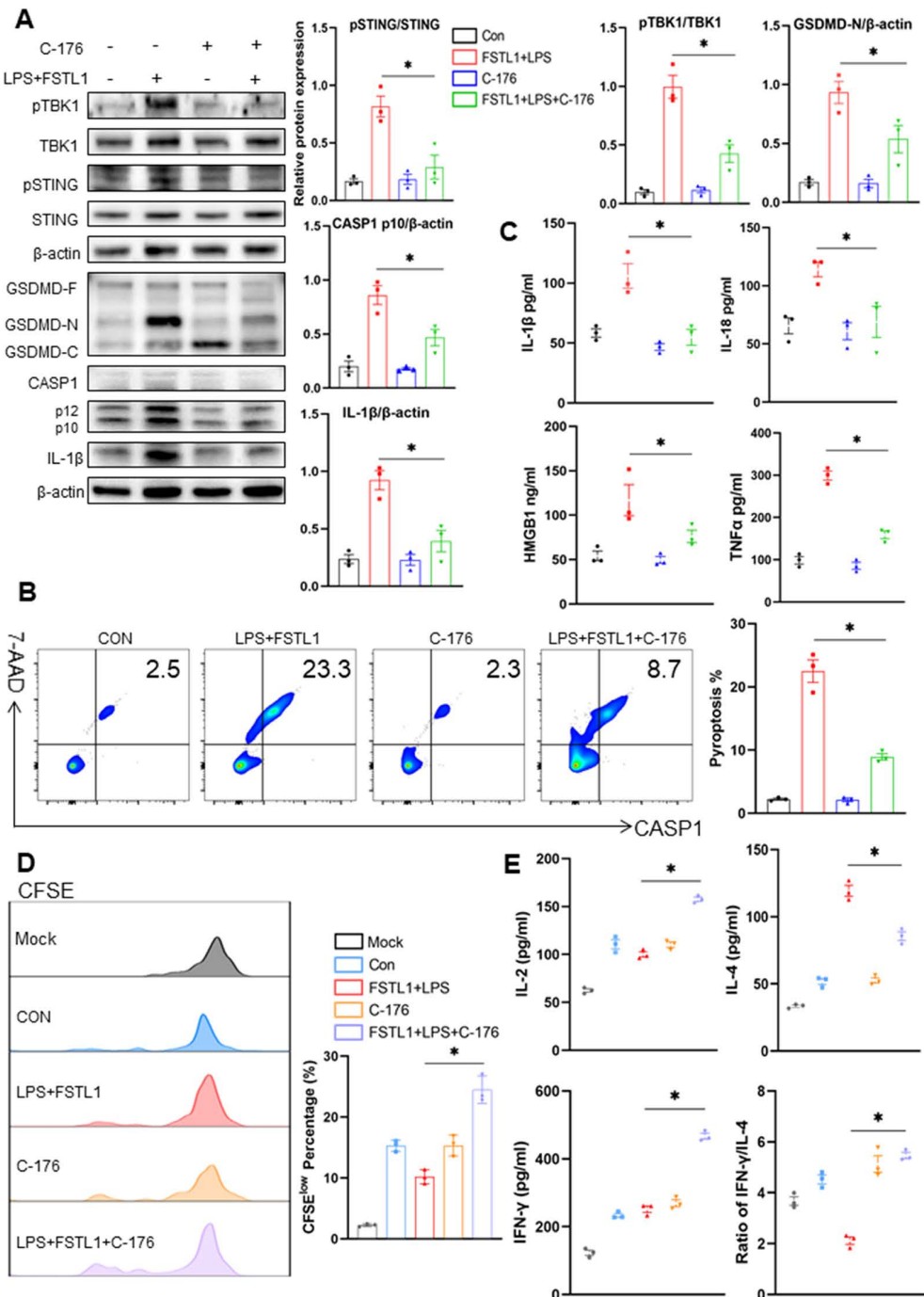

**Fig 4. The inhibition of STING alleviated DC pyroptosis induced by FSTL1. a-c** Splenic CD11+ DCs were isolated from C57BL/6 mice and exposed to the corresponding solvent (control), LPS+FSTL1, C-176 (20 μM), or FSTL1+LPS+C-176 (20 μM) for 24h. The protein levels of cleaved caspase-1, GSDMD, IL-1β, pSTING, STING, pTBK1, TBK1, and β-actin were detected by Western blotting **(a)** (n = 3). The pyroptosis rate of DC was detected using flow cytometry **(b)** (n = 3). The levels of IL-1β, IL-18, HMGB1, and TNF-α in the culture supernatants of DCs were detected using ELISA (c) (n = 3). **d-e** After stimulation, the DCs were co-cultured with T cells. The proliferation of CD4+ T cells co-cultured with DC in each group was measured based on CFSE assay **(d)** (n = 3). The levels of IL-2, IL-4, and IFN-γ and the ratio of IFN-γ/IL-4 in the co-cultured supernatants were detected by ELISA (e) (n = 3). Data were represented as Mean±SD, *P < 0.05.

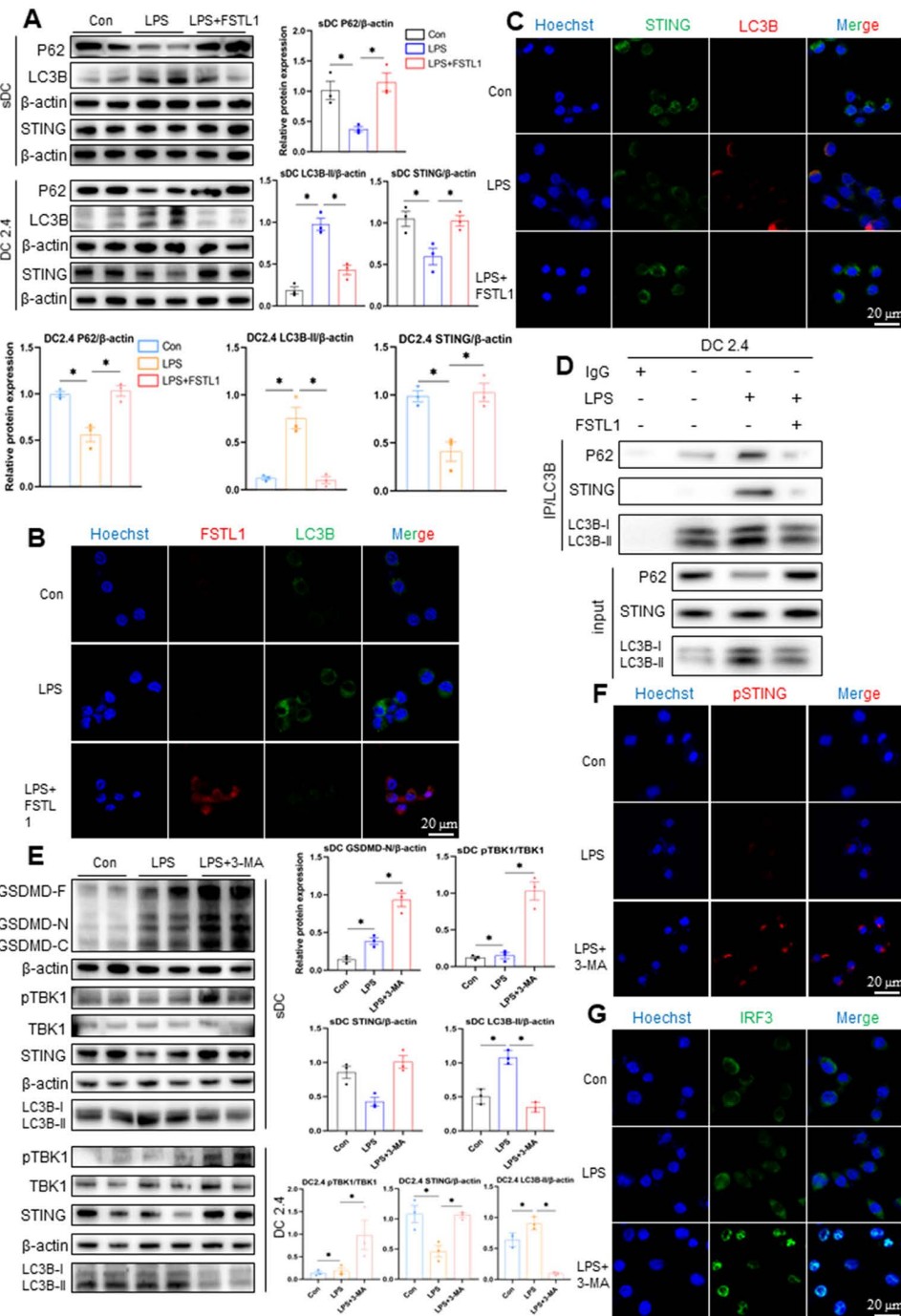

**Fig 5. FSTL1 inhibited the autophagic degradation of STING and promoted STING pathway activation. a-d** Splenic CD11$^+$ DC isolated from C57BL/6 mice and DC2.4 cells cultured in vitro were exposed to corresponding solvent (control), LPS (1 µg/ml), or LPS+FSTL1 (10 ng/ml) for 24h. The protein levels of P62, LC3B, STING, and β-actin were detected by Western blotting **(a)** (n=2). Representative confocal immunofluorescence microscopy images of FSTL1 and LC3B in each DC2.4 group **(b)**. Representative confocal immunofluorescence microscopy images of STING colocalized with LC3B **(c)**. Levels of STING, LC3B, and P62 in the precipitates of the LC3B pull-down from whole cell lysates were evaluated by immunoblotting **(d)**. **e-g** sDCs and DC2.4 cells were treated with vehicle (Con), LPS (1 µg/ml), or LPS combined with the autophagy inhibitor 3-Methyladenine (5 mM) for 24h to mimic the effect of FSTL1. **(e)** (n=2). Western blot analysis of GSDMD-N, pTBK1/TBK1, STING, and LC3B **(f)**. Representative immunofluorescence images of pSTING (red) expression **(g)**. Representative immunofluorescence images showing nuclear translocation of IRF3 (green). Data are represented as Mean ± SD, *P<0.05. Scale bars = 20 µm.

Together, these results demonstrate that FSTL1 disrupts autophagic flux, prevents the autophagic degradation of STING, and thereby potentiates STING-mediated signaling and DC pyroptosis.

### FSTL1 promotes DC pyroptosis and immune dysfunction *in vivo* via a STING-dependent mechanism

Building upon the mechanistic link between FSTL1 and STING-mediated pyroptosis established *in vitro*, we next validated these findings in the sepsis mouse model and evaluate the therapeutic potential of STING inhibition. Mice subjected to CLP were intraperitoneally administered FSTL1 alone or in combination with the STING inhibitor C-176. Compared with the CLP group, mice treated with CLP+FSTL1 showed significant upregulation of GSDMD-N, cleaved CASP1, and IL-1β (Fig 6A). Consistent with this, SYTOX green staining and flow cytometry revealed a substantial increase in the DC pyroptosis rate in the CLP+FSTL1 group (Fig 6B, 6C). These effects were markedly reversed by C-176 treatment, as evidenced by decreased levels of pyroptosis-related proteins and reduced DC pyroptosis in the CLP+FSTL1＋C-176 group (Fig 6A–6C). We further assessed the functional impact of FSTL1 on DC-mediated immune responses. CD4$^+$T cells co-cultured with DCs from the CLP+FSTL1 group exhibited significantly impaired proliferative capacity, which was partially restored by C-176 treatment (Fig 6D). Similarly, co-culture supernatants from the CLP+FSTL1 group contained lower levels of IL-2 and IFN-γ, along with elevated IL-4 and a reduced IFN-γ/IL-4 ratio (Fig 6E). In contrast, inhibition of STING with C-176 enhanced the production of IL-2 and IFN-γ, suppressed IL-4 secretion, and increased the IFN-γ/IL-4 ratio (Fig 6E).

Taken together, these *in vivo* results indicated the central role of the FSTL1-STING axis in regulating DC pyroptosis and adaptive immunity in septic mice.

### Pharmacological inhibition of STING rescues mice from FSTL1-exacerbated sepsis severity and mortality

To further assess the systemic impact of FSTL1 and the therapeutic potential of STING inhibition in sepsis, we evaluated overall immune function, organ injury, and survival in mice. Systemic cytokine profiling in septic mice revealed significantly elevated serum levels of IL-4, IL-10, and TGF-β in the CLP+FSTL1 group compared with the CLP group, accompanied by reduced IL-2 and IFN-γ levels and decreased IFN-γ/IL-4 ratio (Fig 7A). Further evidence of immunosuppression was observed in the CLP+FSTL1 group, which exhibited a pronounced reduction in CD3$^+$CD4$^+$T cells and a marked increase in CD3$^+$CD4$^+$Foxp3$^+$Treg cells (Fig 7B). Notably, administration of C-176 substantially mitigated this FSTL1-induced immune imbalance, restoring the distribution of these T-cell subsets towards that of the sham group (Fig 7C). Furthermore, C-176 treatment attenuated multi-organ damage and improved survival rates in mice receiving CLP+FSTL1＋C-176 compared to those treated with CLP+FSTL1 alone (S3 Fig).

In summary, these findings indicated the critical involvement of FSTL1-triggered STING signaling in worsening sepsis outcomes and suggested that therapeutic targeting of this pathway may alleviate immune dysfunction in sepsis.

## Discussion

In this study, we identified FSTL1 as a novel regulator of DC pyroptosis and immunosuppression in sepsis. We demonstrated that FSTL1 impairs autophagic degradation of STING, leading to its sustained activation and subsequent DC pyroptosis. Both *in vitro* and *in vivo* experiments confirmed that FSTL1 promoted pyroptosis and aggravated immune dysfunction, while inhibition of STING reversed these effects. These findings established the FSTL1-STING axis as a critical mechanism underlying DC depletion in sepsis and highlight its therapeutic potential. Previous studies have suggested that FSTL1 may signal through multiple receptors, including TLR4–CD14 and DIP2A, in different cellular contexts. Although our current results demonstrate that FSTL1 inhibits the autophagic degradation of STING and promotes dendritic cell pyroptosis, the specific upstream receptor mediating this process remains to be elucidated. It is plausible that FSTL1 interacts with one or more of these receptor pathways to influence the STING signaling cascade. Future studies will focus on identifying the receptor responsible for FSTL1-mediated regulation of STING activity and verifying whether this mechanism is conserved across different immune cell populations.

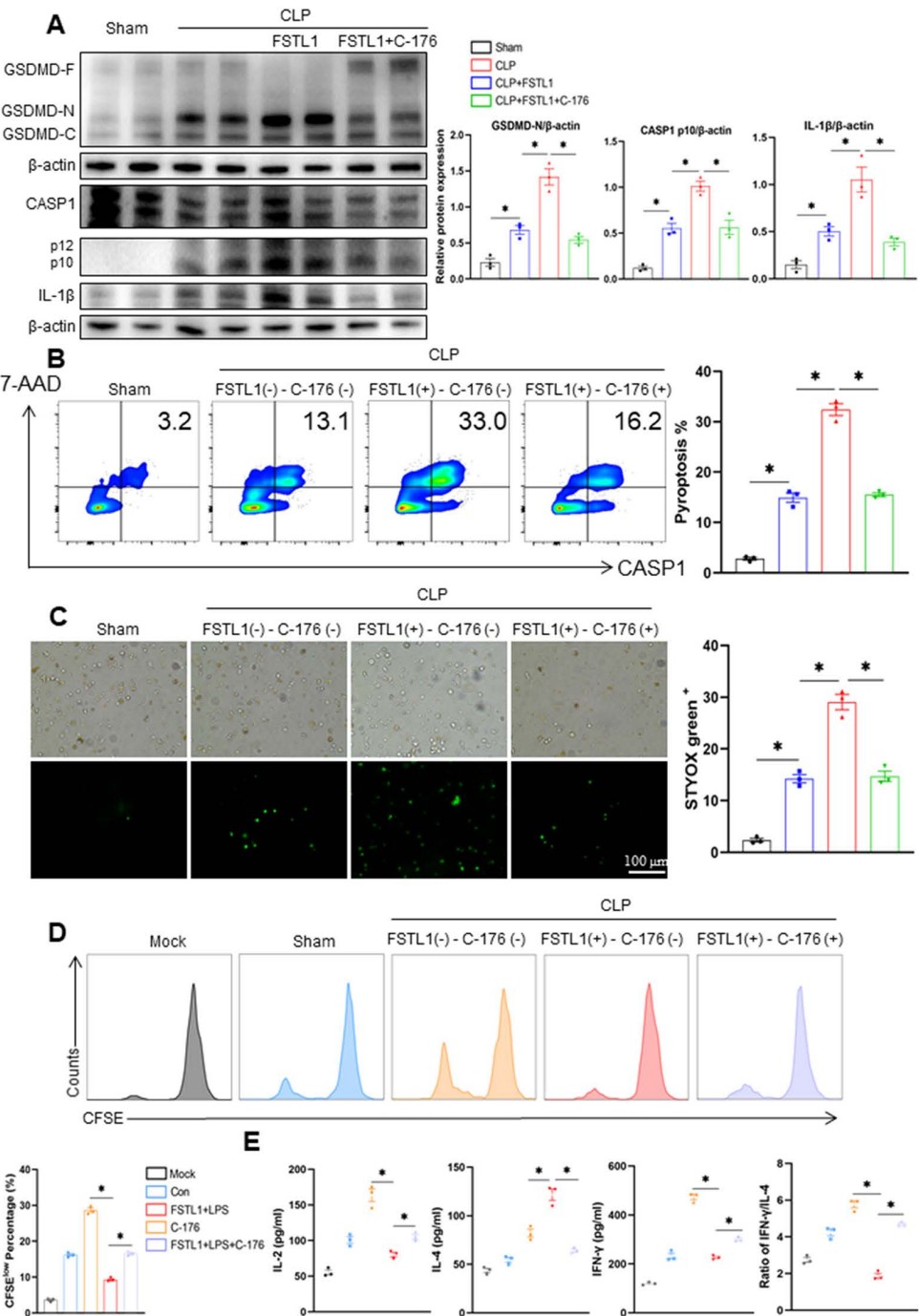

**Fig 6. FSTL1 promoted DC pyroptosis and exacerbated DC immune function in septic mice, which could be partially reversed by C-176. a-d.** C57BL/6 mice were subject to sham or CLP operations and were given corresponding solvent (control), FSTL1 (10 mg/kg), or FSTL1 + C-176 (20 mg/kg) by intraperitoneal injection. Splenic CD11+ DCs were isolated after 24h of the treatment and used for the experiments. The protein levels of GSDMD, cleaved caspase-1, IL-1β, and β-actin detected by Western blotting **(a)** (n = 2). The pyroptosis rate of DC was detected using flow cytometry **(b)** (n = 3). Cell viability was assessed by Sytox green staining **(c)** (n = 3). **d-e** The DCs isolated from experimental mice were co-cultured with T cells. The proliferation of CD4 + T cells co-cultured with DCs in each group were measured based on CFSE assay **(d)** (n = 3). The levels of IL-2, IL-4, and IFN-γ and the ratio of IFN-γ/IL-4 in the co-cultured supernatants were detected by ELISA **(e)** (n = 3). Data were represented as Mean ± SD, *P < 0.05.

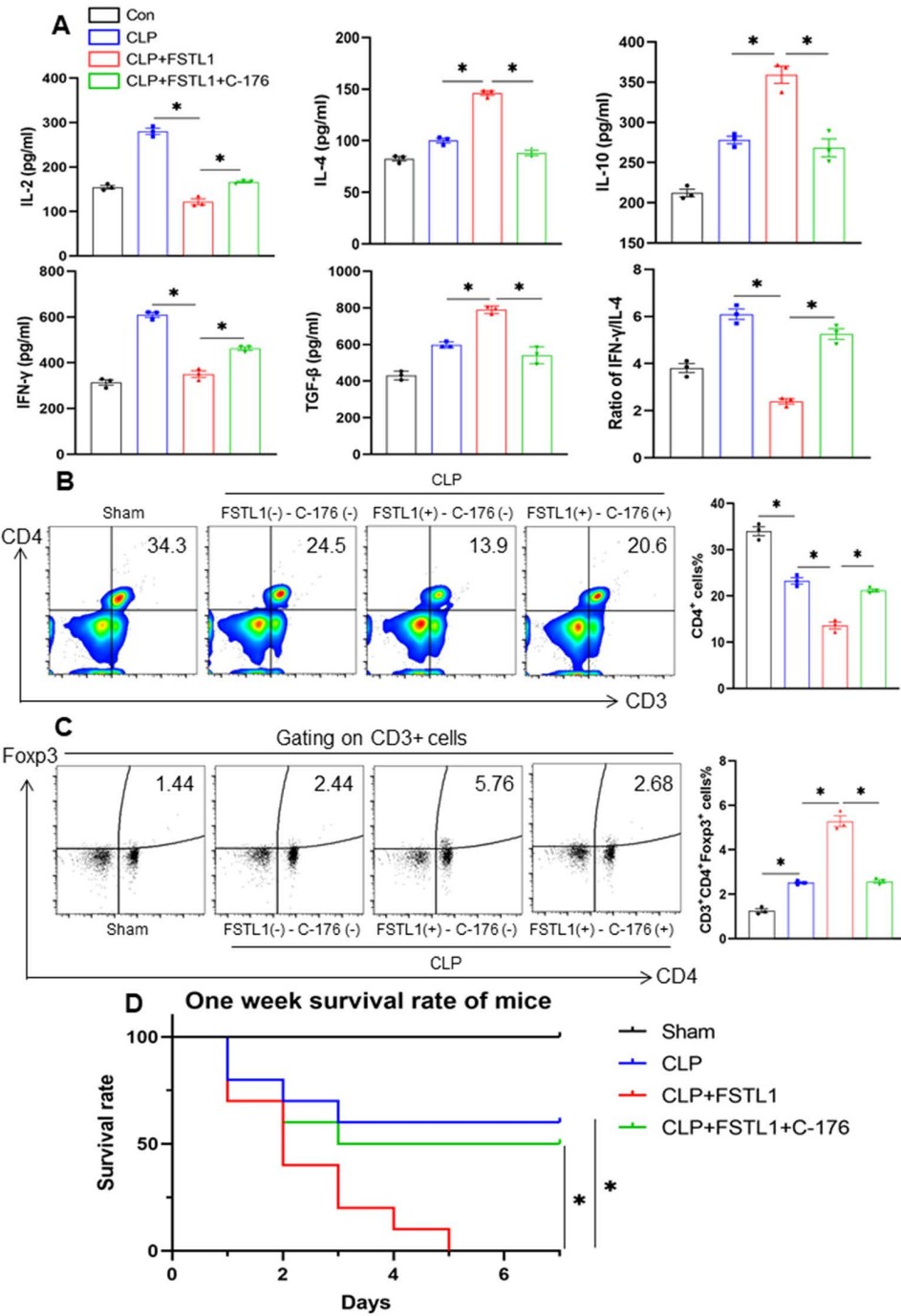

**Fig 7. FSTL1 exaggerated the immunosuppressive state, multiple organ injuries, and mortality in septic mice, which could be partially reversed by C-176. a-d**. C57BL/6 mice were subject to sham or CLP operations and were given corresponding solvent (control), FSTL1 (10 mg/kg), or FSTL1 + C-176 (20 mg/kg) by intraperitoneal injection. Splenic CD11+ DCs were isolated 24h after the treatment and were used for the experiments. The levels of IL-2, IL-4, IL-10, IFN-γ, and TGF-β and the ratio of IFN-γ/IL-4.in mice serum were detected by ELISA **(a)** (n = 3). The proportions of CD3+CD4+ T cells and CD4+Foxp3+ T cells in PBMCs were analyzed by flow cytometry **(b-c)** (n = 3). The survival rates of mice from different groups 7d after CLP **(d)** (n = 6). Data were represented as Mean ± SD, *$P < 0.05$.

FSTL1 has previously been implicated in various inflammatory conditions. For instance, it exacerbates inflammatory responses in chronic obstructive pulmonary disease by modulating autophagy [17], promotes liver fibrosis through metabolic reprogramming in macrophages [18], and contributes to neuroinflammation in models of Alzheimer's disease [19]. Relevant to sepsis, elevated FSTL1 levels have been reported in patient serum, where it enhances NLRP3 inflammasome activation in macrophages [14]. This previously reported pro-inflammatory function of FSTL1 appears to contrast with our finding that it drives immunosuppression. Our study resolves this apparent paradox by proposing a cell-type-specific and context-dependent role for FSTL1. In macrophages, it may initially amplify inflammation. However, in dendritic cells, as we show, it triggers a self-destructive pyroptotic program that ultimately cripples the adaptive immune response, contributing to the later, immunosuppressive phase of sepsis. Our study expands this knowledge by demonstrating a time-dependent increase in serum FSTL1 during sepsis that correlates with immunosuppressive cytokine profiles. We also observed concomitant T cell exhaustion, characterized by reduced CD3$^+$/CD4$^+$ T cells and expanded CD4$^+$Foxp3$^+$ Treg populations. Most importantly, both gain- and loss-of-function experiments showed that FSTL1 promotes DC pyroptosis and exacerbates immune dysfunction, supporting its role as a central mediator of immunosuppression in sepsis.

The STING pathway is a well-established regulator of innate immunity and multiple cell death pathways. Its aberrant activation has been linked to numerous diseases, including sepsis [20,21]. For example, mitochondrial damage and mtDNA release activate cGAS-STING mediated necroptosis in Kupffer cells [22], while STING-dependent apoptosis contributes to intestinal epithelial injury in septic mice [23]. STING promotes pyroptosis through several mechanisms. First, activation of the STING signalling pathway promotes the expression of NLRP3 at the transcriptional level [24]. Secondly, STING directly binds to NLRP3, inhibiting its ubiquitination and degradation [25]. Third, STING mediates the rupture of lysosomes, thus allowing K+ to outflow into lysosomes and promote the activation of NLRP3 [26]. While our study did not dissect the precise downstream mechanism linking STING to GSDMD cleavage in DCs, the observed increase in cleaved CASP-1 suggests the involvement of a canonical inflammasome. Given FSTL1's known connection to NLRP3 [14] and STING's ability to activate it [24–26], it is plausible that FSTL1-potentiated STING signaling converges on an NLRP3-CASP-1 axis to execute pyroptosis in DCs. This hypothesis warrants future investigation. In our study, the STING agonist DMXAA potently induced pyroptosis in DCs. Furthermore, FSTL1 enhanced STING activation, and its pyroptosis-promoting effects were reversed by the STING inhibitor C-176. These results indicate that FSTL1 exacerbates pyroptosis primarily through STING pathway activation. In addition, STING activation is known to induce type I interferon responses. Although IFN-α/β levels were not assessed in the present study, it is possible that type I interferons also contribute to the FSTL1–STING–pyroptosis axis. This hypothesis will be explored in our future investigations.

Autophagy is a conserved physiological process responsible for degrading cytoplasmic components, protein aggregates, and organelles, thereby facilitating metabolic renewal and energy recycling [27]. Based on its functional nature, autophagy is broadly categorized into nonselective and selective types. Nonselective autophagy primarily supports bulk material turnover and energy recovery, whereas selective autophagy involves the targeted degradation of specific substrates via receptor-mediated recognition, playing crucial roles in quality control and the maintenance of cellular homeostasis under various physiological and pathological conditions [27]. Autophagy regulates the STING pathway through multiple mechanisms. For example, P62 interacts with STING and promotes its autophagic degradation after activation [28]. The ubiquitously expressed chaperone UXT enhances the interaction between P62 and STING, further facilitating STING degradation [29]. Additionally, the A137R protein encoded by the African swine fever virus binds to TBK1 and induces its autophagic degradation, thereby suppressing the cGAS-STING pathway [30]. Inhibition of mitophagy also enhances cGAS-STING signaling through mitochondrial DNA release [31,32], while ER-phagy has been shown to target STING for degradation, attenuating STING pathway activation [33]. Our results demonstrate that FSTL1 inhibits autophagic flux and disrupts the STING-LC3B interaction, indicating that FSTL1 enhances STING pathway activation by suppressing its autophagic degradation. This positions FSTL1 as a novel negative regulator of selective autophagy for innate immune sensors. Although our findings demonstrate a strong correlation between FSTL1 expression and suppression of

STING autophagic degradation, the present evidence does not establish direct causation. Further mechanistic studies—such as FSTL1 knockdown or overexpression combined with live-cell imaging of autophagic flux—will be required to clarify the precise role of FSTL1 in autophagy regulation. It is also possible that FSTL1 influences other selective autophagy pathways (e.g., mitophagy, ER-phagy), which merits further exploration. Whether FSTL1 exerts a broader inhibitory effect on other selective autophagy pathways (e.g., mitophagy, xenophagy) in the context of sepsis is an intriguing question. Such a role could imply that elevated FSTL1 might contribute to a more global cellular dysfunction by crippling essential quality control mechanisms, a hallmark of critical illness.

Nonetheless, our study has several limitations. First, although we demonstrated that FSTL1 promotes STING-dependent pyroptosis through autophagy, the precise mechanism by which FSTL1 regulates autophagy remains unclear and warrants further investigation. Although our study focuses on dendritic cells, the potential involvement of other immune cells, including monocytes and macrophages, in FSTL1-mediated pyroptosis cannot be excluded. Future investigations will aim to elucidate the precise mechanisms by which FSTL1 regulates STING degradation and pyroptosis, and to determine whether these regulatory effects extend to additional immune cell populations. Such studies will help establish whether FSTL1 exerts a broader immunomodulatory role in sepsis beyond dendritic cell–specific responses. Second, the detailed molecular events through which STING activation leads to dendritic cell pyroptosis have not been fully elucidated and require additional exploration. Specifically, identifying the precise inflammasome complex (e.g., NLRP3, AIM2) activated downstream of STING in DCs would be a critical next step. Third, future clinical studies are essential to validate these findings and assess their translational potential in human sepsis.

## Conclusion

In conclusion, our findings demonstrate that FSTL1 promotes pyroptosis in DCs by impairing the autophagic degradation of STING, thereby contributing to septic immune dysfunction. These results position FSTL1 as a promising novel biomarker for monitoring immunosuppression and a potential therapeutic target in the treatment of sepsis.

## Materials and methods

### Animals

A total of 48 C57BL/6 mice (male, 6−8 weeks old, 18-22g) were purchased from Beijing HFK Bioscience Co. Ltd. and housed under specific pathogen-free conditions. All experiments were conducted in strict accordance with the National Institutes of Health Guide for the Care and Use of Laboratory Animals and approved by the Scientific Investigation Committee of the PLA General Hospital, Beijing, China (SYXK2016-0014).

### Cecal ligation and puncture

Mice were randomly assigned to four experimental groups using a random number table: (1) sham group, (2) CLP group, (3) CLP+FSTL1 group, and (4) CLP+FSTL1+C-176 group (n=6 per group). After anesthesia via intraperitoneal injection of sodium pentobarbital (80 mg/kg), mice were placed in a supine position, and a ~1 cm midline incision was made to expose the cecum. Approximately 1.0 cm from the cecal end was ligated with a suture. The distal cecum was then punctured once with a 21-gauge needle, and slight pressure was applied to extrude a small amount of fecal content. The cecum was returned to the abdominal cavity, and the incision was closed and disinfected. All mice received subcutaneous injection of 1 mL pre-warmed physiological saline for fluid resuscitation. Mice in the sham group underwent identical surgical procedures including laparotomy and cecal exteriorization, but without ligation or puncture. Mice in the CLP+FSTL1 group received intraperitoneal injection of FSTL1 (10 mg/kg) immediately after CLP surgery. Mice in the CLP+FSTL1+C-176 group received both FSTL1 (10 mg/kg) and C-176 (20 mg/kg) via intraperitoneal injection immediately after CLP. Serum, PBMCs, and splenic DCs were collected at indicated time points (0, 1, 2, and 3 days) for subsequent

analysis. Mice were randomly assigned to four experimental groups using a random number table (n = 6 per group).The sample size was determined based on previous CLP model studies to ensure adequate statistical power to detect a 30% difference in immune and survival parameters ($\alpha = 0.05$, $\beta = 0.2$).

All data collection and histopathological analyses were conducted in a blinded manner.

## Isolation of splenic DCs and PBMCs

Splenic DCs were isolated from splenic single-cell suspensions using a magnetic cell sorting system (Miltenyi Biotech, Bergisch Gladbach, Germany). PBMCs were obtained from whole blood by density gradient centrifugation using a commercial PBMC isolation kit (TBDsciences, Tianjin, China).

## Cell culture and treatment

Splenic DCs obtained from mice were seeded in 6-well plates in complete RPMI 1640 medium supplemented with 10% fetal bovine serum, 100 U/mL penicillin, and 100 µg/mL streptomycin. The DC2.4 cell line (Beijing Qianzhaoxinye Biotechnology Co., Ltd.) was cultured in T25 flasks using DMEM complete medium under the same conditions. All cells were maintained at 37 °C in a humidified atmosphere containing 5% $CO_2$. Cells between passages 3 and 10 were used for all experiments.

For *in vitro* stimulation, cells were treated with LPS (1 µg/mL) alone or in combination with recombinant FSTL1 (10 ng/mL) for 24 h. To inhibit autophagy or the STING pathway, cells were pre-treated for 2 h with 3-methyladenine (3-MA, 5 mM, HY-19312 MCE) or C-176 (20 µM, HY10964 MCE), respectively, prior to LPS and/or FSTL1 stimulation. Control groups received vehicle (PBS or DMSO) treatment alone. After the indicated treatments, cells and supernatants were collected for subsequent analyses.

## Flow cytometry analysis

Cells were harvested and washed twice with cold PBS. For surface marker staining, cells were resuspended in 100 µL of staining buffer and incubated with fluorochrome-conjugated primary antibodies (S1 Table) for 30 minutes at 4 °C in the dark. For intracellular staining, cells were fixed and permeabilized using a commercial fixation/permeabilization kit (Invitrogen) according to the manufacturer's protocol before antibody incubation.

Pyroptosis was assessed by two methods. Early and late-stage cell death was evaluated by co-staining with Annexin V-FITC and 7-AAD according to the manufacturer's instructions (Beijing Pulilai Technology Co., Ltd, E1010). Pyroptotic cells were specifically identified by detecting active Caspase-1 using the FAM-FLICA® Caspase-1 Assay Kit (ImmunoChemistry Technologies, USA) followed by co-staining with a viability dye (7-AAD or DAPI) to gate on membrane-compromised cells (Caspase-1$^+$/7-AAD$^+$).

At least 10,000 events per sample were acquired using a (BD FACSCanto II) flow cytometer (BD Biosciences, CA, USA). Data analysis was performed with FlowJo software (version 10.8.1).

## Western blotting

Proteins extracted from the samples were separated by SDS-polyacrylamide gel electrophoresis (SDS-PAGE) and transferred onto polyvinylidene difluoride (PVDF) membranes. After blocking, the membranes were incubated with primary antibodies (S1 Table) (diluted 1:500–1:1000) at 4 °C overnight. The following day, the membranes were incubated with a horseradish peroxidase (HRP)-conjugated secondary antibody (Goat anti-Rabbit IgG (H + L)-HRP,CST 7074) (diluted 1:10,000) for 60 min at room temperature. Protein bands were visualized using a hypersensitive ECL chemiluminescence kit (Beijing Pulilai Technology Co., Ltd, D2300) and imaged with an ImageQuant LAS 4000 system. The gray values of the target bands were quantified using ImageJ software.

## Immunofluorescence (IF)

Cells grown on coverslips were fixed with 4% paraformaldehyde for 15 minutes... After blocking with 1% BSA, the cells were incubated overnight at 4 °C with the appropriate primary antibody (S1 Table), diluted from 1:50–1:200 in 1% BSA. Following washes, the cells were incubated with a fluorescent secondary antibody (Goat anti-Mouse IgG (H+L) Alexa Fluor 488, CST,4409s]) diluted 1:200 in 1% BSA for 1 hour at room temperature, protected from light. Finally, the cells were washed again and mounted with an anti-fade mounting medium containing DAPI for nuclear counterstaining. Images were acquired using a NIKON CI-S fluorescence microscope equipped with a DS-FI2 imaging system (Nikon, Japan). Confocal images were obtained using a Leica (SP8) laser scanning confocal microscope (Leica, Germany).

## Co-immunoprecipitation (co-IP)

Cells were harvested and pelleted in microcentrifuge tubes, washed twice with PBS, and lysed with RIPA buffer proportional to the cell count. After pre-clearing with protein G agarose beads, the lysates were incubated with the appropriate primary antibody (S1 Table) or control IgG overnight at 4°C with gentle agitation. The immunocomplexes were then captured using protein G agarose beads, followed by five washes with lysis buffer. The immunoprecipitated proteins were eluted and analyzed by immunoblotting using standard procedures.

## Enzyme-linked immunosorbent assay (ELISA)

The concentrations of target proteins in cell culture supernatants and murine serum were quantified using commercial ELISA kits according to the manufacturer's instructions. The specific kits were as follows: IL-2 (Shanghai Enzyme-linked Biotechnology Co., Ltd,ml002095), IL-4 (Shanghai Enzyme-linked Biotechnology Co., Ltd, ml002097), IL-10 (Shanghai Enzyme-linked Biotechnology Co., Ltd., ml002100), TGF-β (Shanghai Enzyme-linked Biotechnology Co., Ltd., ml037870), IL-1β (Shanghai Enzyme-linked Biotechnology Co., Ltd., ml002093), IL-18 (Shanghai Enzyme-linked Biotechnology Co., Ltd., ml037887), HMGB1 (Shanghai Enzyme-linked Biotechnology Co., Ltd., ml037885), TNF-α (Shanghai Enzyme-linked Biotechnology Co., Ltd., ml002023), and IFN-γ (Shanghai Enzyme-linked Biotechnology Co., Ltd., ml002096). Absorbance was measured at 450 nm using a microplate reader.

## Statistical analysis

Data are presented as mean±SD. Statistical analyses were performed using GraphPad Prism (version 9.4.1) and SPSS (version 22.0).Comparisons between two groups were performed using an unpaired two-tailed Student's t-test, while multiple-group comparisons were analyzed using one-way ANOVA followed by Tukey's post hoc test.$P < 0.05$ was considered statistically significant.The number of biological replicates (n) for each experiment is indicated in the corresponding figure legends.

## Supporting information

**S1 Fig. The purity and pyroptosis analysis of splenic DCs from mice.** (**a**) The purity analysis of the splenic CD11+ DCs isolated from mice. (**b**) The pyroptosis rate of DC was calculated using flow cytometry. Data were represented as Mean±SD, *$P < 0.05$.
(TIF)

**S2 Fig. The co-immunoprecipitation of LC3B, P62, and STING in DC2.4 after 3-MA treatment.** DC2.4 cultured *in vitro* were exposed to the corresponding solvent (control), LPS, or LPS+3-MA for 24h. LC3B was immunoprecipitated from whole cell lysates of DC2.4, and levels of STING, LC3B, and P62 in the precipitates were evaluated by immunoblotting.
(TIF)

**S3 Fig. TUNEL staining of multiple organ injuries in septic mice.** C57BL/6 mice were subject to sham or CLP operations and were given corresponding solvent (control), FSTL1 (10 mg/kg), FSTL1 + C-176 (20 mg/kg) by intraperitoneal injection. Kidney, liver, lung and intestine were collected 24 hours post-treatment and subjected to TUNEL staining.
(TIF)

**S4 Fig. Dose-dependent effect of recombinant FSTL1 on GSDMD cleavage in splenic DCs.** Splenic dendritic cells were treated with recombinant FSTL1 at 5, 10, or 20 µg/mL for 24 hours (n = 2). Western blot analysis was performed to detect the expression of GSDMD-Full and GSDMD-N-terminal. FSTL1 at 10 µg/mL showed the highest level of GSDMD cleavage.
(TIF)

**S5 Fig. Serum FSTL1 concentrations across different experimental mouse groups.** C57BL/6 mice were divided into four groups: Sham, CLP, FSTL1 (10 mg/kg), and CLP + FSTL1 (10 mg/kg) (n = 3). Serum FSTL1 concentrations were measured by ELISA 24 hours after treatment. FSTL1 levels were increased in septic mice compared with sham controls, moderately elevated in the FSTL1 group, and further increased in the CLP + FSTL1 group.
(TIF)

**S1 Table. List of antibodies.**
(DOCX)

## Author contributions

**Conceptualization:** Qiong Li, Xingui Dai.

**Data curation:** Qiong Li, Hua Ling, Jinru Li.

**Formal analysis:** Qiong Li, Hua Ling, Jinru Li.

**Funding acquisition:** Qiong Li, Lijun Guo, Xingui Dai.

**Investigation:** Qiong Li, Hua Ling, Jingyi Li, Jinru Li, Wentao Duan, Lijun Guo.

**Methodology:** Qiong Li, Hua Ling, Jingyi Li, Jinru Li, Wentao Duan, Lijun Guo.

**Project administration:** Qiong Li, Lijun Guo, Xingui Dai.

**Resources:** Hua Ling, Jingyi Li, Jinru Li, Wentao Duan.

**Software:** Qiong Li, Hua Ling, Jingyi Li, Jinru Li, Lijun Guo.

**Supervision:** Lijun Guo, Xingui Dai.

**Validation:** Qiong Li, Jingyi Li, Wentao Duan.

**Visualization:** Qiong Li, Wentao Duan.

**Writing – original draft:** Qiong Li, Jingyi Li.

**Writing – review & editing:** Qiong Li, Lijun Guo, Xingui Dai.

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
