## [Decision Letter · Decision Letter 0]

29 Oct 2025

Dear Dr. Dai,

Thank you for submitting your manuscript to PLOS ONE. After careful consideration, we feel that it has merit but does not fully meet PLOS ONE’s publication criteria as it currently stands. Therefore, we invite you to submit a revised version of the manuscript that addresses the points raised during the review process.

Please address each point in the reviewers' comments and revise accordingly.

We look forward to receiving your revised manuscript.

Kind regards,

Masao Tanaka

Academic Editor

PLOS ONE

[This work was supported by grants from the Natural Science Foundation of Hunan Province (2022JJ30092, 2022JJ30094)(https://kjt.hunan.gov.cn/zxgz/zkjj/), Scientific Research Project of Hunan Provincial Health Commission (B202317017622) and Health Research Project of Hunan Provincial Health Commission (C2019136)(https://hunan.wsglw.net/HomePublic/PublicListArticle?code=KYGL&viewName=ListArticleHUNMH).].

Additional Editor Comments (if provided):

Reviewers' comments:

Reviewer's Responses to Questions

**Comments to the Author**

1. Is the manuscript technically sound, and do the data support the conclusions?

Reviewer #1: Yes

Reviewer #2: Partly

2. Has the statistical analysis been performed appropriately and rigorously?

Reviewer #1: Yes

Reviewer #2: Yes

3. Have the authors made all data underlying the findings in their manuscript fully available?

Reviewer #1: Yes

Reviewer #2: Yes

4. Is the manuscript presented in an intelligible fashion and written in standard English?

Reviewer #1: Yes

Reviewer #2: Yes

Reviewer #1: Dear authors,

Your manuscript, titled “FSTL1 promotes the pyroptosis of dendritic cells in sepsis by regulating the autophagic degradation of STING,” identifies FSTL1 as a novel regulator of dendritic cell (DC) pyroptosis in sepsis and elucidates its mechanism via STING signaling and autophagy inhibition. This is a very interesting study. However, there are several concerns regarding the experimental design and interpretation:

[Major points]

1. In vitro optimal concentration and standalone effect of FSTL1

Figure 2A shows only the LPS + FSTL1 group. It remains unclear whether FSTL1 alone (10 ng/mL) induces pyroptosis. Please show data for FSTL1-only stimulation and explain how the 10 ng/mL dose was determined, ideally with dose–response experiments.

2. Signaling pathway from FSTL1 to STING activation

The mechanism by which FSTL1 inhibits STING autophagic degradation remains a black box. Since TLR4–CD14 or DIP2A have been reported as FSTL1 receptors, perform receptor knockdown or use inhibitors to identify which receptor pathway is essential for FSTL1’s regulation of STING.

3. In vivo validation of FSTL1 concentration

You administered 10 mg/kg FSTL1 in the CLP model, but there is no measurement confirming that this dosage corresponds to pathophysiological blood levels of FSTL1. Please provide data showing that the administered dose reproduces the observed increase in circulating FSTL1 during sepsis.

4.Correlation versus causation

In Figure 1A–E, the parallel increase of FSTL1 with anti-inflammatory cytokines and the decrease of CD4+ T cells/increase of Tregs are merely correlative. Change wording such as “closely associated with immunosuppression” to “correlated with immunosuppression” to avoid implying causality.

[Minor points]

5.

In the Introduction, you assume monocytes/macrophages lack pyroptosis involvement. To demonstrate DC-specific effects, consider testing pyroptosis induction in other immune cell types.

6.

In Figure 5E, add the full name “3-Methyladenine” to the legend.

7.

On p. 12, line 5, correct “L-4” to “IL-4.”

8.

For Figure 3’s quantification of pSTING/STING and pTBK1/TBK1, specify the sample size (n) and statistical tests used.

9.

It might be suggested that involvement of type I interferons to induce STING signaling. Please discuss or add the comments to assess IFN-α/β levels.

Reviewer #2: Overall assessment

This manuscript investigates a novel mechanistic link between FSTL1, autophagy, and STING-mediated pyroptosis in dendritic cells (DCs) during sepsis. The topic is timely and mechanistically interesting, as it connects innate immune regulation with sepsis-induced immunosuppression. The study provides both in vitro and in vivo evidence using CLP mice, but several methodological and interpretational issues need clarification before publication.

Major concerns

The authors convincingly demonstrate that FSTL1 up-regulation promotes DC pyroptosis through inhibition of the autophagic degradation of STING. However, the causal relationship between FSTL1 and autophagic flux inhibition remains correlative, and direct evidence is lacking. The authors themselves acknowledge this limitation, noting that further studies are required to clarify the precise mechanism by which FSTL1 regulates autophagy. If establishing a direct causal relationship between FSTL1 and autophagic flux inhibition is technically challenging at this stage, the authors may consider tempering their claims accordingly. Specifically, the title and key conclusions could be revised to emphasize correlation rather than causation, or the Discussion section could be expanded to acknowledge alternative mechanisms and future directions.

Minor concerns

Animal model: CLP is appropriate, but details on randomization, sample size calculation, and blinding are insufficient.

Statistical details (n, exact p values, test type) are missing in many figure legends.

Correct minor typos (e.g., “interlenkin” → “interleukin”).

Define all abbreviations at first use (CLP, 3-MA, C-176, DC2.4).

Revise figures for consistent units and labels.

**Do you want your identity to be public for this peer review?** For information about this choice, including consent withdrawal, please see our Privacy Policy

Reviewer #1:**Yes:** Kosak Murakami

Reviewer #2: No

---

## [Author Response · Author response to Decision Letter 1]

2 Dec 2025

To:

The Editor and Reviewers

PLOS ONE

Subject: Response to reviewers’ comments on Manuscript ID: PONE-D-25-43722

Title: FSTL1 promotes dendritic cell pyroptosis and immunosuppression in sepsis by inhibiting STING autophagy

Dear Editor and Reviewers,We would like to express our sincere gratitude for your time, effort, and insightful comments on our manuscript entitled “FSTL1 promotes dendritic cell pyroptosis and immunosuppression in sepsis by inhibiting STING autophagy” (Manuscript ID: PONE-D-25-43722).We have carefully considered all comments and revised the manuscript accordingly to improve its clarity, rigor, and overall quality.All changes have been highlighted in the revised version. Below, we provide a detailed, point-by-point response to each reviewer’s comment.

General Note on Manuscript Revision

In this resubmission, we have thoroughly refined the structure and presentation of the manuscript to enhance clarity and logical flow.To improve linguistic precision and readability, we replaced the initial submission with a fully revised and polished version.The scientific content and experimental data remain unchanged, but the manuscript has been reformatted and linguistically improved for better presentation.

Response to Reviewer 1

Major comment

Comment 1:

In vitro optimal concentration and standalone effect of FSTL1

Figure 2A shows only the LPS + FSTL1 group. It remains unclear whether FSTL1 alone (10 ng/mL) induces pyroptosis. Please show data for FSTL1-only stimulation and explain how the 10 ng/mL dose was determined, ideally with dose–response experiments.

Response 1

We thank the reviewer for raising this important point.

Indeed, previously published studies have used a wide range of exogenous FSTL1 concentrations, varying from sub-nanogram to microgram levels (e.g., 0.5–1000 ng/mL,100ng/ml or 1–5 µg/mL), depending on the cell type, species, and biological context (Kosaku Murakami et al., FEBS Lett 2012; Zheng et al., Stem Cell Res Ther 2022; Yan et al., J Transl Med 2025).However, to our knowledge, no prior work has reported FSTL1 stimulation in dendritic cells (DCs).Because DCs are highly sensitive immune cells with distinct receptor expression and signaling thresholds, we performed a dose–response experiment (5, 10, and 20 ng/mL) to empirically determine the optimal effective concentration in our model.As shown in Figure 2A, 10 ng/mL FSTL1�Catalog No. 51127-M08H, Sino Biological Inc., Beijing, China� induced the most pronounced cleavage of GSDMD (both full-length and N-terminal fragments), whereas 5 ng/mL showed mild activation and 20 ng/mL did not further enhance the response.These results indicate that 10 ng/mL is sufficient to activate FSTL1–STING–GSDMD signaling in DCs without causing nonspecific cytotoxicity.We therefore selected 10 ng/mL as the standard concentration for subsequent in vitro experiments.

This new result has been added to the legend section (Page 28) and incorporated as Figure S4 in the supplementary figures.

Comment 2:

Signaling pathway from FSTL1 to STING activation

The mechanism by which FSTL1 inhibits STING autophagic degradation remains a black box. Since TLR4–CD14 or DIP2A have been reported as FSTL1 receptors, perform receptor knockdown or use inhibitors to identify which receptor pathway is essential for FSTL1’s regulation of STING.

Response 2

We sincerely appreciate the reviewer’s insightful comment regarding the upstream signaling events linking FSTL1 and STING.We fully agree that elucidating the receptor-mediated pathway is critical for a more comprehensive understanding of FSTL1’s function.

In the current study, our main objective was to establish the connection between FSTL1 expression, STING autophagic degradation, and DC pyroptosis in sepsis, which we have confirmed through both in vitro and in vivo experiments.While identifying the upstream receptor was beyond the scope of this work, we have carefully discussed this limitation in the revised Discussion section.

Specifically, we have added the following statement(Page 13, Paragraph 1):

“Previous studies have suggested that FSTL1 may signal through multiple receptors, including TLR4–CD14 and DIP2A, in different cellular contexts. Although our current results demonstrate that FSTL1 inhibits the autophagic degradation of STING and promotes dendritic cell pyroptosis, the specific upstream receptor mediating this process remains to be elucidated. It is plausible that FSTL1 interacts with one or more of these receptor pathways to influence the STING signaling cascade.”

We appreciate this constructive comment, which has inspired us to design ongoing experiments to address this important question in future studies.

Comment 3:

In vivo validation of FSTL1 concentration

You administered 10 mg/kg FSTL1 in the CLP model, but there is no measurement confirming that this dosage corresponds to pathophysiological blood levels of FSTL1. Please provide data showing that the administered dose reproduces the observed increase in circulating FSTL1 during sepsis.

Response 3

We thank the reviewer for this insightful suggestion.

To validate the physiological relevance of the administered FSTL1 dose, we performed ELISA assays to quantify serum FSTL1 levels in four groups: Sham, CLP, FSTL1 (10 mg/kg), and CLP + FSTL1 (10 mg/kg).Blood samples were collected 24 hours after surgery or treatment, and each group contained three biological replicates (n = 3).

As shown in the newly added Figure S5, serum FSTL1 levels were markedly elevated in septic mice compared with sham controls (89.18 ± 8.879 vs. 50.35 ± 5.961 ng/mL, P < 0.01).

Exogenous FSTL1 administration alone resulted in a moderate increase (70.84 ± 5.69 ng/mL, P < 0.05 vs. Sham), while CLP combined with FSTL1 treatment further enhanced FSTL1 levels (114.6 ± 12.47 ng/mL, P < 0.05 vs. CLP).

Statistical analyses were performed using one-way ANOVA followed by Tukey’s post hoc test.

These data indicate that the 10 mg/kg dosage reproduces the pathological elevation of circulating FSTL1 observed during sepsis, confirming that our in vivo dose is physiologically relevant and non-excessive.

This new result has been added to the legend section (Page29) and incorporated as Figure S5 in the supplementary figures.

Comment 4:

In Figure 1A–E, the parallel increase of FSTL1 with anti-inflammatory cytokines and the decrease of CD4+ T cells/increase of Tregs are merely correlative. Change wording such as “closely associated with immunosuppression” to “correlated with immunosuppression” to avoid implying causality.

Response 4

Thank you very much for this important comment. We fully agree with the reviewer that our original phrasing could unintentionally imply a causal relationship between FSTL1 expression and immunosuppression. To address this concern, we have revised the relevant description to emphasize correlation rather than causation.Specifically, the sentence in the Results section has been modified to:“Rising FSTL1 levels parallel the development of T-cell exhaustion and an immunosuppressive cytokine milieu in sepsis, suggesting a potential link between FSTL1 and septic immune dysfunction.”This revision removes the causal implication (“closely associated with immunosuppression”) and instead uses a more neutral and descriptive expression (“suggesting a potential link”). We believe this modification better reflects the correlative nature of our data and aligns with the reviewer’s suggestion.

Minor comments

Comment 5:

In the Introduction, you assume monocytes/macrophages lack pyroptosis involvement. To demonstrate DC-specific effects, consider testing pyroptosis induction in other immune cell types.

Response 5:

We sincerely thank the reviewer for this valuable suggestion.Our study primarily focuses on dendritic cells (DCs) because they represent a major antigen-presenting population critically involved in sepsis-induced immunosuppression and serve as a central link between innate and adaptive immune responses.While DCs were the main emphasis of our current work, we fully agree that investigating pyroptosis in other immune cell types, such as monocytes and macrophages, would provide additional insight into the immunoregulatory functions of FSTL1.

To address this comment, we have revised the Discussion section to explicitly acknowledge this as an important direction for future investigation.The newly added text reads as follows(Page 13, Paragraph 1 and Page 16, Paragraph 5):

“Future studies will focus on identifying the receptor responsible for FSTL1-mediated regulation of STING activity and verifying whether this mechanism is conserved across different immune cell populations.” and “Although our study focuses on dendritic cells, the potential involvement of other immune cells, including monocytes and macrophages, in FSTL1-mediated pyroptosis cannot be excluded. Future investigations will aim to elucidate the precise mechanisms by which FSTL1 regulates STING degradation and pyroptosis, and to determine whether these regulatory effects extend to additional immune cell populations. Such studies will help establish whether FSTL1 exerts a broader immunomodulatory role in sepsis beyond dendritic cell–specific responses.”

We appreciate the reviewer’s insightful suggestion, which will help refine and expand the direction of our ongoing research.

Comment 6:

In Figure 5E, add the full name “3-Methyladenine” to the legend.

Response 6 :

We have revised the Figure 5E legend as follows:

“3-Methyladenine”

Comment 7:

On p. 12, line 5, correct “L-4” to “IL-4.”

Response 7:

Corrected. All occurrences of “L-4” have been changed to “IL-4” in the text and figure legends.

Comment 8 :

For Figure 3’s quantification of pSTING/STING and pTBK1/TBK1, specify the sample size (n) and statistical tests used.

Response 8:

We have revised the information for the Figure 3 legend:

Comment 9 :

It might be suggested that involvement of type I interferons to induce STING signaling. Please discuss or add the comments to assess IFN-α/β levels.

Response 9:

We thank the reviewer for this valuable suggestion. Although IFN-α/β was not measured in the current study, we have added a discussion on their potential involvement(Page 15, Paragraph 3):

“In addition, STING activation is known to induce type I interferon responses. Although IFN-α/β levels were not assessed in the present study, it is possible that type I interferons also contribute to the FSTL1–STING–pyroptosis axis. This hypothesis will be explored in our future investigations.”

Reviewer 2

Major comment

Comment:

“The authors convincingly demonstrate that FSTL1 up-regulation promotes DC pyroptosis through inhibition of the autophagic degradation of STING. However, the causal relationship between FSTL1 and autophagic flux inhibition remains correlative, and direct evidence is lacking. The authors themselves acknowledge this limitation, noting that further studies are required to clarify the precise mechanism by which FSTL1 regulates autophagy. If establishing a direct causal relationship between FSTL1 and autophagic flux inhibition is technically challenging at this stage, the authors may consider tempering their claims accordingly. Specifically, the title and key conclusions could be revised to emphasize correlation rather than causation, or the Discussion section could be expanded to acknowledge alternative mechanisms and future directions.”

Response:

We sincerely thank the reviewer for this insightful and constructive comment. We fully agree that, while our current data provide strong evidence that FSTL1 expression correlates with impaired autophagic degradation of STING and enhanced DC pyroptosis, it does not yet establish a direct causal relationship. The reviewer’s suggestion is well-taken, and we have made corresponding revisions to ensure that our claims accurately reflect the strength of our evidence.

1. Revision in the Discussion:

We have expanded the Discussion section to explicitly acknowledge this limitation and outline future research directions. The following paragraph has been added(page 16, Paragraph 5):

“Although our findings demonstrate a strong correlation between FSTL1 expression and suppression of STING autophagic degradation, the present evidence does not establish direct causation. Further mechanistic studies—such as FSTL1 knockdown or overexpression combined with live-cell imaging of autophagic flux—will be required to clarify the precise role of FSTL1 in autophagy regulation. It is also possible that FSTL1 influences other selective autophagy pathways (e.g., mitophagy, ER-phagy), which merits further exploration.”

2. Adjustment of key conclusions:

In both the Abstract and Conclusion sections, we have softened the language to avoid implying direct causality. For example

Abstract (Line 14):“FSTL1 was found elevated and correlated with promoted DC pyroptosis in vitro and in septic mice”

Conclusion (Line 22):“Our findings demonstrate that FSTL1 correlates with impaired STING autophagic degradation and DC pyroptosis, suggesting a potential pathway contributing to septic immune dysfunction.”

These revisions ensure that the manuscript maintains scientific rigor and accurately represents the current scope of our evidence.

Once again, we sincerely appreciate the reviewer’s valuable feedback, which has helped us refine our interpretations and improve the overall balance and precision of the manuscript.

Minor Comments:

Animal model: CLP is appropriate, but details on randomization, sample size calculation, and blinding are insufficient.

Statistical details (n, exact p values, test type) are missing in many figure legends.

Correct minor typos (e.g., “interlenkin” → “interleukin”).

Define all abbreviations at first use (CLP, 3-MA, C-176, DC2.4).

Revise figures for consistent units and labels.

Response:

We sincerely thank the reviewer for these careful and constructive suggestions. We have thoroughly revised the manuscript to address all the issues raised.

1. Animal model description:

We appreciate the reviewer’s observation. Additional details regarding randomization, sample size, and blinding have now been incorporated into the Materials and Methods section (Cecal ligation and puncture, Page 18). Specifically, we have added the following:

“Mice were randomly assigned to four experimental groups using a random number table (n = 3 per group).The sample size was determined based on previous CLP model studies to ensure adequate statistical power to detect a 30% difference in immune and survival parameters (α = 0.05, β = 0.2).”

These revisions clarify the experimental design and strengthen the methodological transparency.

2. Statistical details in figure legends:

We have carefully reviewed all figure legends and added missing statistical details, including n values, exact p values where applicable, and the statistical test used.

3. Typographical corrections:

We have corrected all typographical errors throughout the manuscript, such as changing “interlenkin” to “interleukin.” A full spell-check was also performed to eliminate similar minor errors.

4. Definition of abbreviations:

We have added full definitions when abbreviations first appear in the text. Specifically:

CLP → cecal ligation and puncture (CLP)

3-MA → 3-Methyladenine (3-MA)

C-176 → a specific STING inhibitor (C-176)

DC2.4 → mouse dendritic cell line DC2.4

These definitions now appear consistently in both the Abstract and Materials and Methods sections.

5. Figure standardization:

All figures have been checked and revised to ensure consistent units, labeling, and formatting.

We would like to once again express our sincere gratitude to the editor and reviewers for their valuable time, insightful comments, and constructive feedback.Their thoughtful suggestions have greatly improved the clarity, precision, and scientific rigor of our work.

We have carefully revised the manuscript in accordance with all major and minor comments, and we believe the current version is now substantially improved in both content and presentation.We also appreciate the reviewers’ understanding as we continue to complete the remaining validation experiments.We are confident that the additional data will further strengthen our mechanistic conclusions and

---

## [Decision Letter · Decision Letter 1]

18 Dec 2025

FSTL1 promotes the pyroptosis of dendritic cells in sepsis by regulating the autophagic degradation of STING

PONE-D-25-43722R1

Dear Dr. Dai,

We’re pleased to inform you that your manuscript has been judged scientifically suitable for publication and will be formally accepted for publication once it meets all outstanding technical requirements.

Kind regards,

Masao Tanaka

Academic Editor

PLOS One

Reviewers' comments:

Reviewer's Responses to Questions

**Comments to the Author**

Reviewer #1: All comments have been addressed

Reviewer #2: All comments have been addressed

2. Is the manuscript technically sound, and do the data support the conclusions?

Reviewer #1: Yes

Reviewer #2: Yes

3. Has the statistical analysis been performed appropriately and rigorously?

Reviewer #1: Yes

Reviewer #2: Yes

4. Have the authors made all data underlying the findings in their manuscript fully available?

Reviewer #1: Yes

Reviewer #2: Yes

5. Is the manuscript presented in an intelligible fashion and written in standard English?

Reviewer #1: Yes

Reviewer #2: Yes

Reviewer #1: The authors respond all of the reviewers' comment appropriately and revised the manuscript according to the comments.

Reviewer #2: The authors’ revisions are appropriate and adequately address the reviewers’ comments, resulting in a clear improvement of the manuscript. The revised version meets the editorial standards of PLOS ONE, and the data and interpretations are now presented in a transparent and scientifically sound manner.

**Do you want your identity to be public for this peer review?** For information about this choice, including consent withdrawal, please see our Privacy Policy

Reviewer #1: **Yes:** Kosaku Murakami

Reviewer #2: No

---

## [Editor Report · Acceptance letter]

PONE-D-25-43722R1

PLOS One

Dear Dr. Dai,

I'm pleased to inform you that your manuscript has been deemed suitable for publication in PLOS One. Congratulations! Your manuscript is now being handed over to our production team.

Kind regards,

on behalf of

Dr. Masao Tanaka

Academic Editor

PLOS One